# Tissue-resident macrophages can be generated *de novo* in adult human skin from resident progenitor cells during substance P-mediated neurogenic inflammation *ex vivo*

**Jennifer Gherardini**[1], **Youhei Uchida**[2], **Jonathan A. Hardman**[3], **Jérémy Chéret**[4], **Kimberly Mace**[5], **Marta Bertolini**[1©], **Ralf Paus**[1,3,4©] *

**1** Monasterium Laboratory GmbH, Münster, Germany, **2** Department of Dermatology, Kagoshima University Graduate School of Medical and Dental Sciences, Kagoshima, Japan, **3** Centre for Dermatology Research and NIHR Manchester Biomedical Research Centre University of Manchester, Manchester, United Kingdom, **4** Dr. Phillip Frost Department of Dermatology and Cutaneous Surgery, University of Miami Miller School of Medicine, Miami, Florida, United States of America, **5** Division of Cell Matrix Biology and Regenerative Medicine, University of Manchester, Manchester, United Kingdom

© These authors contributed equally to this work.

* ralf.paus@manchester.ac.uk

**Data Availability Statement:** All relevant data are within the paper and its Supporting Information files.

## Abstract

Besides monocyte (MO)-derived macrophages (MACs), self-renewing tissue-resident macrophages (trMACs) maintain the intracutaneous MAC pool in murine skin. Here, we have asked whether the same phenomenon occurs in human skin using organ-cultured, full-thickness skin detached from blood circulation and bone marrow. Skin stimulation *ex vivo* with the neuropeptide substance P (SP), mimicking neurogenic skin inflammation, significantly increased the number of CD68⁺MACs in the papillary dermis without altering intracutaneous MAC proliferation or apoptosis. Since intraluminal CD14⁺MOs were undetectable in the non-perfused dermal vasculature, new MACs must have differentiated from resident intracutaneous progenitor cells in human skin. Interestingly, CD68⁺MACs were often seen in direct cell-cell-contact with cells expressing both, the hematopoietic stem cell marker CD34 and SP receptor (neurokinin-1 receptor [NK1R]). These cell-cell contacts and CD34⁺cell proliferation were up-regulated in SP-treated skin samples. Collectively, our study provides the first evidence that resident MAC progenitors, from which mature MACs can rapidly differentiate within the tissue, do exist in normal adult human skin. That these NK1R⁺trMAC-progenitor cells quickly respond to a key stress-associated neuroinflammatory stimulus suggests that this may satisfy increased local MAC demand under conditions of wounding/ stress.

## Introduction

Macrophages (MACs) are mononuclear phagocytic leukocytes that play a key role in adaptive and innate immunity, and regulate tissue homeostasis [1–4]. While long believed to derive

**Funding:** This work was initiated and mainly performed while RP, JG, MB, JC, YU were affiliated to the University of Münster and continued when the authors moved their affiliations to Monasterium Laboratory, and/or University of Miami. The work was partially supported by a basic research grant and a PhD fellowship (salary) awarded to JG from Monasterium Laboratory GmbH, Münster, as well as by the NIHR Manchester Biomedical Research Centre, and by University of Miami start-up funds. RP is the founder and CEO of Monasterium laboratory, while JG, MB, JC are or were employed of Monasterium Laboratory GmbH. The funders had no role in data collection and analysis. The specific roles of these authors are articulated in the 'author contributions' section.

**Competing interests:** The authors have read the journal's policy and the authors of this manuscript have the following competing interest: MB and JC are or were employed of Monasterium Laboratory GmbH. JG is a paid employee of Monasterium Laboratory GmbH, Münster. RP is the founder and CEO of Monasterium laboratory. This does not alter our adherence to PLOS ONE policies on sharing data and materials.

from circulating monocytes (MOs) [5–7], in most examined adult murine tissues, including skin, MACs are entirely or partially self-maintained from proliferating tissue-resident MACs (trMACs) of embryonal origin [8–11]. Moreover, during tissue inflammation, the contribution of MOs to the increase of MAC number is minimal and is due in large part to the proliferation of trMACs in murine tissues [10,12–14].

However, our current understanding of MAC ontogeny and differentiation in peripheral tissues largely relies on studies in mice and remains unclear whether these concepts are transferable to the human system, namely to human skin. Yet, the fact that patients with congenital monocytopenia still have skin MACs [15,16] supports the hypothesis that the pool of MACs in human skin is either self-maintained or generated by locally resident progenitor cells. Interestingly, it has already been demonstrated for human skin and upper airway mucosal mast cells, that they can mature from resident progenitor cells [17–19], and can be expanded in the absence of circulating progenitors, and bone marrow derived-stem cells.

Therefore, the current pilot study aimed to clarify whether, as in mice, the dermal MAC pool in adult human skin is self-maintained and can be expanded in the absence of hemoperfusion with circulating MOs and bone marrow derived-stem cells.

To address it, full-thickness hair-bearing human skin fragments were organ-cultured detached from blood circulation and bone marrow under serum-free conditions [20,21] and compared MAC number and activities in both a steady-state and pro-inflammatory conditions. For the latter, we simulated neurogenic inflammation through the administration of the prototypic stress-associated sensory neuropeptide, substance P (SP) [22], which acts primarily via neurokinin-1 receptor (NK1R) and Mas-related G Protein coupled receptor X2 (MRGPRX2) [23] and is a key mediator of neurogenic skin inflammation [22,24–26]. This design was also chosen because intracutaneous SP administration increases the number of intradermal MACs in several rodent models *in vivo* [24,25]. The number, proliferation and apoptosis of CD68$^+$MACs [27,28] and of putative MAC precursors, namely of CD34$^+$cells [29,30], was assessed in human dermis by quantitative (immuno-)histomorphometry [31]. Finally, preliminary mechanistic experiments were performed using the specific NK1R antagonist, aprepitant [32–34], in order to clarify how SP triggers the de novo generation of MAC in human skin.

## Materials and methods

### Human tissue collection and full-thickness skin organ culture

All experiments on human tissue were performed according to Helsinki guidelines. As a laboratory that specializes in hair research with special interest in the role of perifollicular macrophages in scalp skin, we purposely used healthy frontotemporal human hairy scalp skin samples from women undergoing cosmetic facelift surgery, obtained from collaborating plastic surgeons, after written patient consent and ethics committee approval from the University of Münster (n. 2015-602-f-S), which severely limited the amount of available human skin for organ culture. 4mm skin fragments were obtained from the skin samples upon arrival to the laboratory after overnight shipment, and organ cultured as previously described [20,35] with minor modifications.

To better conserve the viability of immunocytes, a mixture of William's E and RPMI medium (1:1), which contains insulin, hydrocortisone and L-glutamine [20,21] was used.

After a 24h of equilibration period, skin punches were treated with $10^{-8}$, $10^{-10}$ M of SP or with a corresponding vehicle control (media only).

Alternatively, before and during SP stimulation, the selective NK1R antagonist, aprepitant [32–34] was administered at $10^{-7}$M, in order to prevent the effect of SP.

To test DNA synthesis, samples were treated for 24h with 10µM EdU (5-ethynyl-2'-deox-yuridine) [36].

To test if the endothelial cells can be activated by SP, skin biopsies were incubated either with $10^{-10}$ M of SP or with 2 different concentrations (0.5–50 ng/ml) of TNFα for 24h [21]. (for details, see S1 Text).

### Immunohistochemistry/Immunofluorescence microscopy and quantitative (immuno-)histomorphometry

After acetone fixation, skin cryo-sections were incubated with the primary antibodies listed in S1 Table over night at 4˚C, or 1h at 37˚C, after appropriate pre-incubation with serum (S2 Table) and with the appropriate secondary antibody (S3 Table) or solutions provided by staining kits (for details, see S1 Text).

The number of single or double-positive cells were evaluated by quantitative (immuno-)histomorphometry [31] in the papillary dermis, in an area defined as 200µm from the basement membrane of the epidermis (S1 Fig), or in the whole skin section using Biozero-II Analyzer software (for details, see S1 Text).

### Statistics

Data are expressed as number, percentage or fold change over vehicle or day 0 when vehicle was not determined. All data were analysed with GraphPad Prism 6 software (GraphPad Prism). Statistical significance was calculated by One-way ANOVA test for parametric data, or Kruskal-Wallis test for non-parametric data. Bonferroni's test or Dunn's test were used, respectively, as post hoc test. $p < 0.05$ was regarded as significant.

## Results and discussion

### SP selectively increases the number of resident CD68[+]MACs in human papillary dermis *ex vivo*

After 24h of SP stimulation ($10^{-10}$M and $10^{-8}$M [21]) *ex vivo*, the number of cells expressing CD68, a late endosomal glycoprotein which selectively demarcates human MACs [27,28], was significantly increased in human papillary dermis compared to control samples (Fig 1).

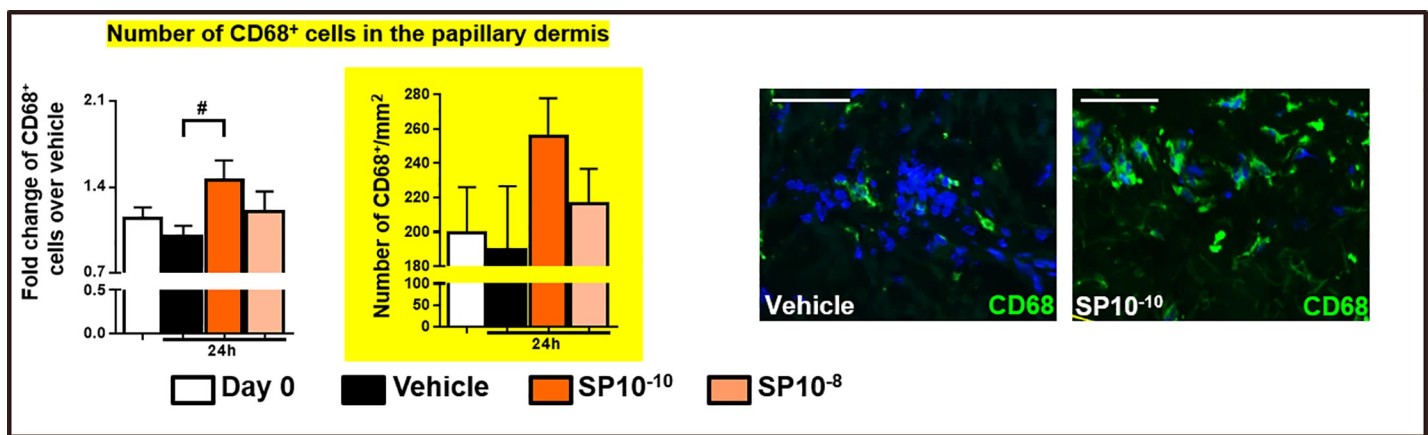

**Fig 1. Effect of SP on CD68[+]cells.** Quantitative analysis and representative images of CD68[+]cells in human skin fragments at day 0 or treated with vehicle, or SP *ex vivo*. The number of positive cells was counted in the papillary dermis (200µm from the epidermis). N = 15–16 skin biopsies/group from 4 different donors. Fold change of Mean or Mean ± SEM. One-way ANOVA, post hoc test Bonferroni (#p<0.05). Scale bare: 50µm.

We detected a difference of CD68$^+$cell number between samples treated with SP $10^{-10}$M and vehicle samples of 66.8 cell/mm$^2$ (i.e. an increase of 34% compared to vehicle controls) (Fig 1). Given that the increase in the number of CD68$^+$MACs consistently occurred within only 24 h of SP stimulation, and this in a non-blood-perfused tissue, this evidence constitutes a remarkable numeric enhancement.

Instead, the overall number of antigen-presenting cells in human skin, including MACs positive for MHC class II [37,38], remained essentially unchanged (S2A Fig).

This unexpected result rose the hypothesis that the increase number of CD68$^+$MAC may be counteracted by the depletion of other MHCII$^+$ cells, namely dendritic cells.

It is known, in fact, that under conditions of neurogenic inflammation, dermal dendritic cells fast respond and migrate to the lymph node [24,39]. Given that several subtypes of dendritic cells are present in human dermis and that it is still not entirely clear which is the best marker [40,41], we have opted for a double immunostaining protocol of MHCII with CD11c. Our results showed that the number of MHCII$^+$CD11c$^+$ dendritic cells is reduced by ca. 20% in SP treated samples as compared to vehicle samples (S2B Fig). This explains the essentially unaltered number of MHCII$^+$cells after SP stimulation, since the decrease in CD11c$^+$cells is counterbalanced by the observed increase in the number of (also MHCII$^+$) CD68$^+$MACs. Thus, intradermal MACs in human skin and/or their progenitor cells are highly responsive to SP stimulation, even in the absence of functional sensory skin innervation.

## MOs are unlikely to significantly contribute to the *de novo* generation of CD68$^+$MACs in human skin *ex vivo*

The newly formed dermal CD68$^+$MACs could plausibly have derived from circulating MOs. However, when we have investigated the number of MOs trapped in the lumen of blood vessels by using double-immunostaining to visualize both the endothelium (CD31$^+$cells) and CD14$^+$MOs, hardly any CD14$^+$MOs were found to be trapped in CD31$^+$blood vessels (Fig 2A).

Furthermore, the number of MO-derived MACs (i.e. CD14$^+$CD68$^+$cells) [42], which represented ca. 75% of the total MAC population found in human skin *ex vivo* (Fig 2B), was unaffected by SP treatment (Fig 2B). Interestingly, Tamoutounour and colleagues, demonstrate that in mouse skin around 20% of CD68$^+$MACs are Ly-6C- (i.e. the mouse analogue of CD14), indicating the existence of a pool of dermal MACs that is established prenatally and persists in adulthood, independently from circulating monocytes [10,42,43], showing that MACs are indeed "partially" self-maintained by proliferating tissue-resident MACs of embryonal origin.

Therefore, the fact that we also found around 25% of CD68$^+$MACs in human skin to be negative for CD14 nicely correlates with mouse data and suggests that a substantial portion of human dermal MACs are also maintained independently from circulating monocytes.

In addition, H&E staining was also used to investigate whether blood was still trapped in capillaries after skin processing and organ culture, and to discriminate macrophage (large irregularly shaped cells) from monocytes (smaller and more rounded cells) [43,44] in the capillaries.

We could not find any MACs trapped in the blood vessels of any skin samples analysed (S3 Fig), confirming that the isolate CD14$^+$cells found in capillaries were indeed CD14$^+$ circulating MOs. We have also found very few histochemically stained red blood cells in skin samples at day 0 but not in vehicle or SP treated skin samples, i.e. organ cultured samples (S3 Fig). Therefore, while some blood remained trapped in the blood vessels after punches preparation, most of it was washed out during organ culture, further supporting our hypothesis that the newly generated MAC after SP stimulation did not derive from CD14$^+$ circulating monocytes.

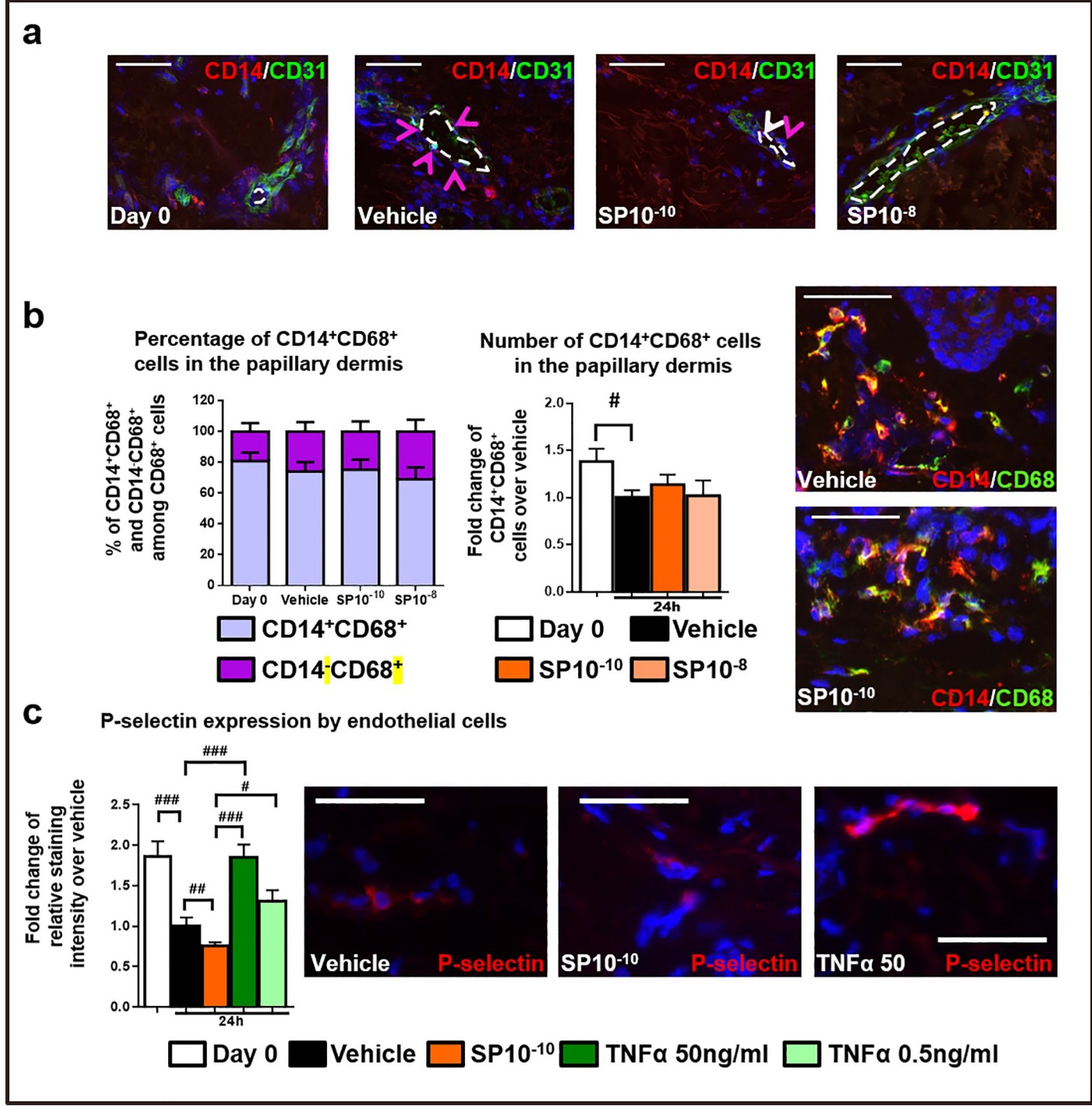

**Fig 2. CD14+ MOs in human skin *ex vivo*.** (a) Representative images of CD14/CD31cells in human skin fragments at day 0 or treated with vehicle, or SP ex vivo. The CD14+cells were visualized in the entire dermis. N = 8 skin biopsies/group from 2 different donors (126 skin sections). Arrows indicate CD14+MOs trapped in the lumen of blood vessel cells (white arrows), and CD14+MOs localized close to blood vessels (violet arrows). (b) Quantitative analysis, and representative images of CD14/CD68 in immunofluorescence staining in human skin fragments at day 0 or treated with vehicle, or SP *ex vivo*. The percentage and number of double-positive cells was counted in the papillary dermis (200μm from the epidermis). N = 8 skin biopsies/group from 2 different donors. Mean ± SEM, One-way ANOVA, post hoc test Bonferroni (# p<0.05). (c) Quantitative analysis, and representative images of P-selectin in immunofluorescence staining in human skin fragments at day 0 or treated with vehicle, SP or TNFα *ex vivo*. Staining intensity was evaluated in the entire dermis. N = 8 skin biopsies/group from 2 different donors. Fold change of Mean ± SEM, Kruskal-Wallis-Test, post hoc test Dunn, #p<0.05, ##p<0.01, ###p<0.001. Scale bare 50μm.

Moreover, MOs extravasation is strictly regulated via protein-protein interaction with endothelial cells [45]. In this process, several adhesion proteins are involved in the capturing and rolling phase, namely P-selectin [46,47]. P-selectin is a transmembrane lectin, whose expression is up-regulated by different cytokines, including TNFα [47].

To test if the endothelial cells are activated and overexpress P-selectin after SP stimulation, we treated the skin biopsies either with SP ($10^{-10}$M) or with 2 different concentrations (0.5–50 ng/ml) of TNFα [21]. As reported, P-selectin was strongly up-regulated in CD31[+]cells in response to TNFα [47] (Fig 2C) indicating that endothelial cells can be activated *ex vivo*. Instead, SP did not promote P-selectin expression on them (Fig 2C).

Of interest, the fact that P-selectin expression is up-regulated in endothelial cells [45,46] of day 0 skin samples as compared to vehicle samples may well result from the pro-inflammatory environment triggered in the skin sample during the manipulation of the skin samples, from the trauma of surgical skin harvesting to the initiation of the culture (day 0) [48]. Thus, the expression of P-selectin is most likely restored to the baseline level during organ culture, as the skin gradually adjusts to its new ex vivo environment (Fig 2C).

This renders very unlikely that SP enhances the capacity of the extremely few CD14[+]MOs trapped in intracutaneous blood vessel *ex vivo* to extravasate in a P-selectin-dependent manner.

Therefore, together with the absence of functional blood flow in our assay system, the fact that SP did not activate the endothelial cells (Fig 2C) and the extremely rare presence of CD14[+]cells in human skin blood vessels (Fig 2A) virtually rules out extravasating MOs as a credible source of the substantial, SP-induced increase in the number of dermal CD68[+]cells.

## The SP-induced MAC increase cannot be explained by suppression of MAC apoptosis or stimulation of MAC proliferation

Next, we checked whether the SP-induced MAC increase resulted from a protective effect of SP on dermal CD68[+]MACs apoptosis under *ex vivo* conditions. On the contrary, the percentage of apoptotic (TUNEL[+] or active-caspase-3[+]) CD68[+]MACs was significantly up-regulated in SP-treated skin (Fig 3A), suggesting that the SP-induced increase in the intradermal MAC count was even higher than Fig 1 indicates. This also demonstrates that SP promotes human dermal MAC apoptosis within their physiological environment.

When the expression of proliferation-associated parameters (Ki-67, M-phase-specific phospho-histone 3 [PH3]), and DNA synthesis using EdU incorporation [49]) by CD68[+]MACs were evaluated, no significant differences were seen between vehicle- and SP-treated skin (Fig 3B; S4 Fig).

Moreover, MHCII is a molecule expressed in several mature cell populations in human skin in the dermis, namely antigen presenting cells (CD14[+] and CD14[-] dendritic cells, and MACs), and non-antigen presenting cells [50]. We have already essentially excluded that CD14[+]cells and CD68[+]MACs, which also express MHCII, in the dermis as a credible source for the SP-induced increased in the number of dermal CD68[+]cells. Therefore, the only other MHCII[+]cell subtype we did not investigate that could give rise to MACs, are CD14[-] dendritic cells [51].

We have, therefore, determined the number of proliferative MHCII[+]cells in the papillary dermis. However, the percentage of proliferative MHCII[+]cells is significantly down-regulated either in vehicle or SP treated groups compared to freshly isolated skin (Fig 3C). The decrease in the proliferative MHCII[+] cell number may be explained by the well-documented fact that MHCII[+] dendritic cells, rather than macrophages [52], rapidly migrate out of the skin into the

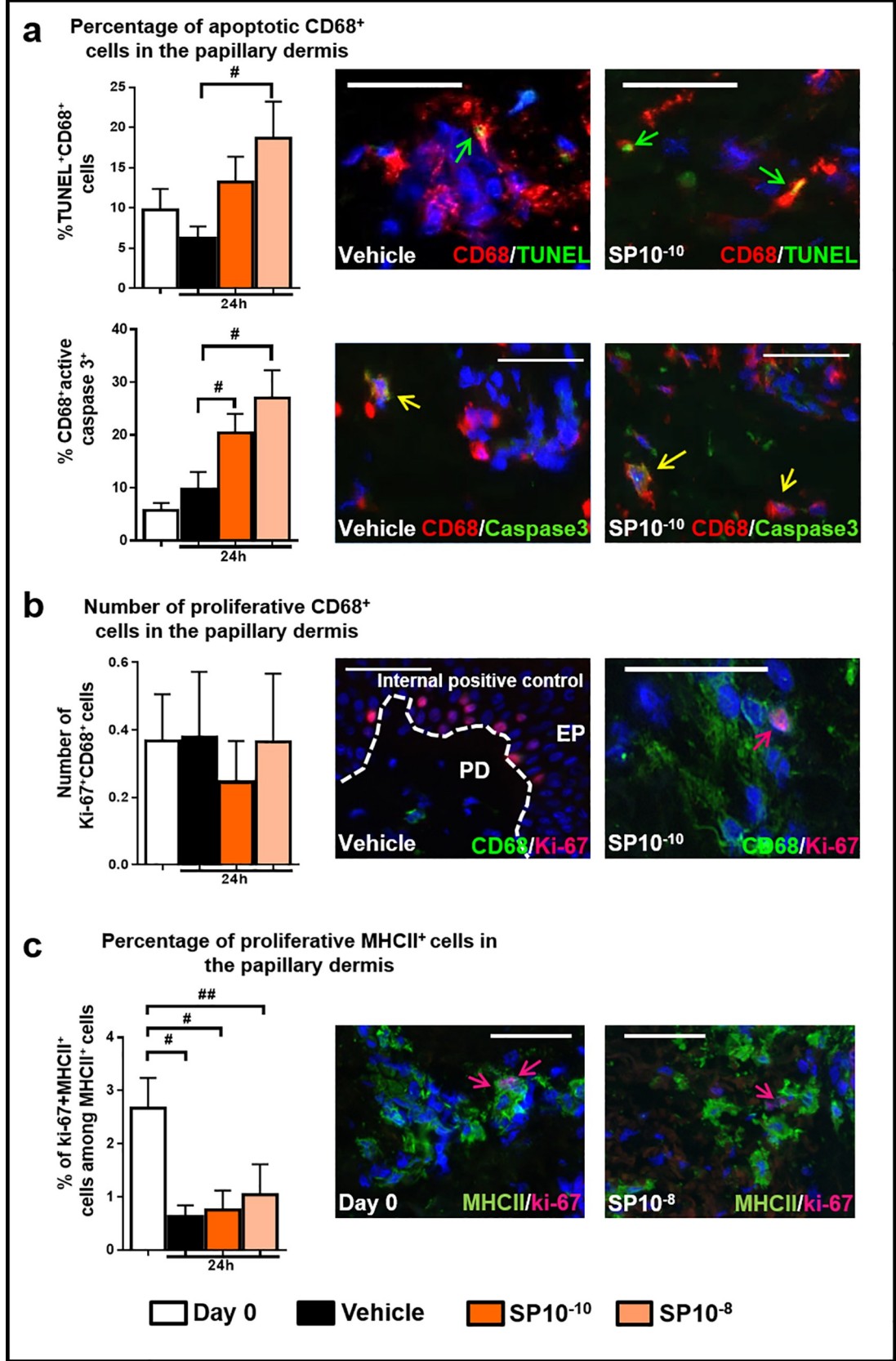

**Fig 3. Proliferative and apoptotic MACs.** (a) Quantitative analysis and representative images of CD68/TUNEL or CD68/Caspase3 cells in human skin fragments at day 0 or treated with vehicle, or SP *ex vivo*. The percentage of double-positive cells was

counted in the papillary dermis (200μm from the epidermis). N = 15–16 skin biopsies/group from 4 different donors (CD68/ TUNEL) or N = 7–8 skin biopsies/group from 2 different donors (CD68/Caspase3). Mean ± SEM. One-way ANOVA, post hoc test Bonferroni (#p<0.05), Kruskal-Wallis-Test, post hoc test Dunn, (#p<0.05). Arrows indicate CD68+TUNEL+cells (green arrows), CD68+active-caspase3+ (yellow arrows). (b) Quantitative analysis and representative images of CD68/Ki-67 cells in human skin fragments at day 0 or treated with vehicle, or SP ex vivo. Representative picture of the internal positive control for Ki-67+proliferative cells. The percentage of double-positive cells was counted in the entire dermis. N = 15–16 skin biopsies/group from 4 different donors. Mean ± SEM. Kruskal-Wallis-Test, post hoc test Dunn, ns. Arrow indicates CD68+Ki-67+cells (red). (c) Quantitative analysis, and representative images of MHCII+/Ki-67+cells in immunofluorescence staining in human skin fragments at day 0 or treated with vehicle, or SP ex vivo. The percentage of double-positive cells was counted in the papillary dermis (200μm from the epidermis). N = 11–12 skin biopsies/group from 3 different donors. Fold change of Mean ± SEM, One-way ANOVA, post hoc test Bonferroni (#p<0.05, ##p<0.01). Arrow indicates MHCII+Ki-67+cells (pink). EP: epidermis; PD: papillary dermis. Scale bare: 50μm.

culture medium under tissue stress conditions [24,39]. This explain also the downregulation of MHCII+CD11c+ in S2B Fig.

This renders highly unlikely that any of the MHCII+cell populations in the dermis is a source of the increased number of CD68+cells.

Thus, the observed SP-induced increase in the number of CD68+MACs cannot be credibly explained by apoptosis-protection or the proliferation/self-renewal of CD68+MACs or other MHCII+cells.

## Resident CD34+-progenitor cells in human dermis are the most likely source of intracutaneously generated CD68+MACs

These results strongly suggested that adult human dermis harbours immature, resident progenitors, from which mature CD68+MACs can rapidly be differentiated *in loco*, e.g. from previously deposited mesenchymal and/or hematopoietic stem cells (MSC, HSC). Immunohistology for classical MSC and/or HSC markers (c-Kit, CD34) [53,54] showed that almost all c-Kit+cells in the papillary dermis were CD68- and phenotypically represented mast cells [17,31], and that their number did not differ between vehicle- and SP-treated samples (Fig 4A). Instead, very rarely, CD68+CD34+cells could be seen in the papillary dermis of vehicle and SP-treated skin (Fig 4B), many of which expressed NK1R (S5A Fig) and were in direct physical contact with CD68+CD34-cells; the latter phenomenon notably increased after SP treatment (Fig 4C). Since endothelial cells can also express CD34 (S5B Fig) it is important to note that almost 40% of dermal CD34+cells were CD31- (S5B Fig), and thus likely represented CD34+HSCs (S5B Fig).

## SP promotes the proliferation of CD34+ dermal progenitor cells and their maturation into CD68+MACs

This in turn invited the hypothesis that the SP-induced increase in CD68+MACs (Fig 1) is primarily brought about by impacting on resident progenitor cells, from which these skin MACs differentiate. In fact, quantitative analysis of CD34/Ki-67 double-immunohistochemistry revealed that SP significantly upregulated the number of proliferating (i.e. Ki-67+) CD34+cells (Fig 5A), showing that SP can stimulate the proliferation of human dermal CD34+cells within their natural tissue niche.

Moreover EdU-incorporating CD34+cells were seen in close contact with EdU-incorporating CD68+cells (Fig 5B).

## Direct activation of NK-1 receptor on CD34+MAC progenitors is involved in SP-induced MAC de novo generation *ex vivo*

Given that CD34+cells express NK1R (S5A Fig), and that the increase in MAC numbers occurs quite rapidly (within 24 hrs) after SP stimulation, our data support the hypothesis that SP

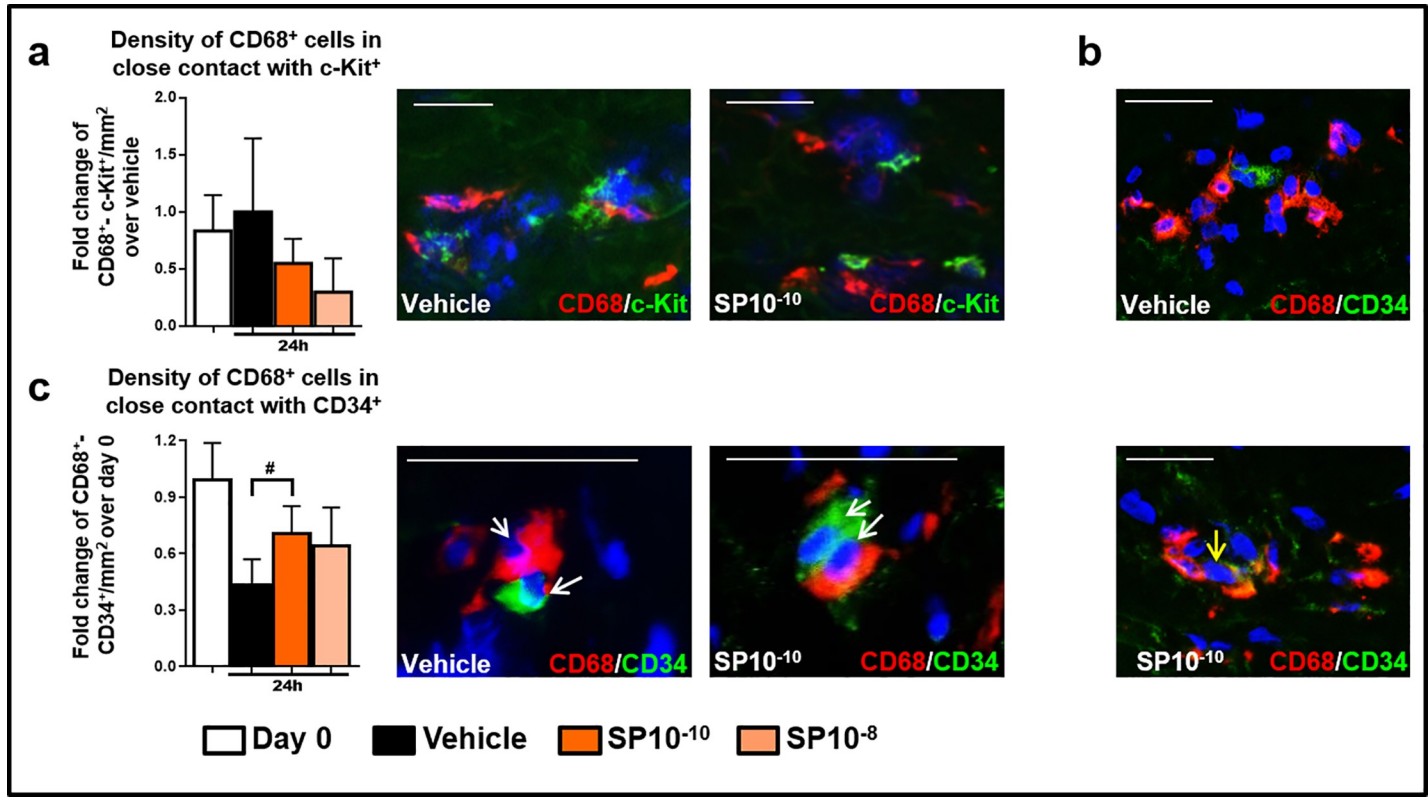

**Fig 4. Identification of c-Kit⁺ cells and CD34⁺ progenitors in human skin.** (a) Quantitative analysis, and representative images of CD68/c-Kit in immunofluorescence staining in human skin fragments at day 0 or treated with vehicle, or SP *ex vivo*. The number of cells in close contact was counted in the papillary dermis (200μm from the epidermis). N = 15–16 skin biopsies/group from 4 different donors. Fold change of Mean ± SEM. Kruskal-Wallis-Test, ns, post hoc test Dunn, ns. (b) Representative pictures of CD34⁺ and CD68⁺ cells. The double positive cells were visualized in the papillary dermis (200μm from the epidermis). Yellow arrow indicates double positive CD68⁺CD34⁺ cell. (c) Quantitative analysis, and representative images of CD68/CD34. The number of cells in close contact was counted in the papillary dermis (200μm from the epidermis). N = 15–16 skin biopsies/group from 4 different donors. Fold change of Mean ± SEM. Kruskal-Wallis-Test, post hoc test Dunn (#p<0.05). Arrows indicate CD68⁺MACs in close contact with CD34⁺cells (white). Scale bare: 50μm.

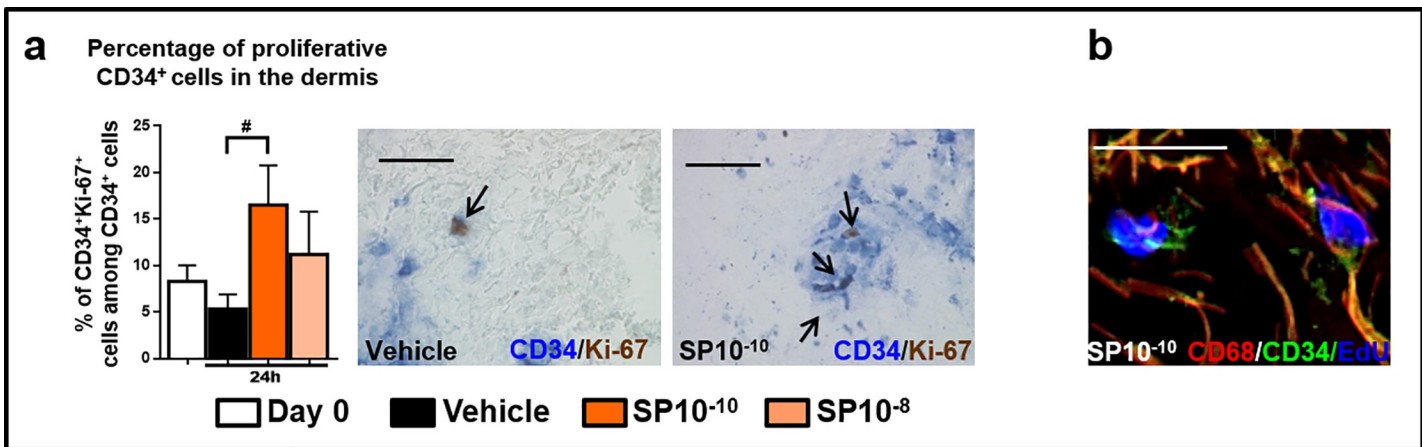

**Fig 5. Proliferative CD34⁺progenitor cells.** (a) Quantitative analysis, and representative images of CD34/Ki-67. The number of double positive cells was counted in the papillary dermis (200μm from the epidermis). N = 15–16 skin biopsies/group from 4 different donors. Mean ± SEM. Kruskal-Wallis Test, post hoc test Dunn (#p<0.5). Black arrows indicate CD34⁺Ki-67⁺cells. (b) Representative pictures of CD68⁺CD34⁺EdU⁺cells. The double positive cells (CD68⁺EdU⁺ or CD34⁺EdU⁺) were visualized in the papillary dermis. N = 12 skin biopsies/group from 3 different donors. Scale bare: 50μm.

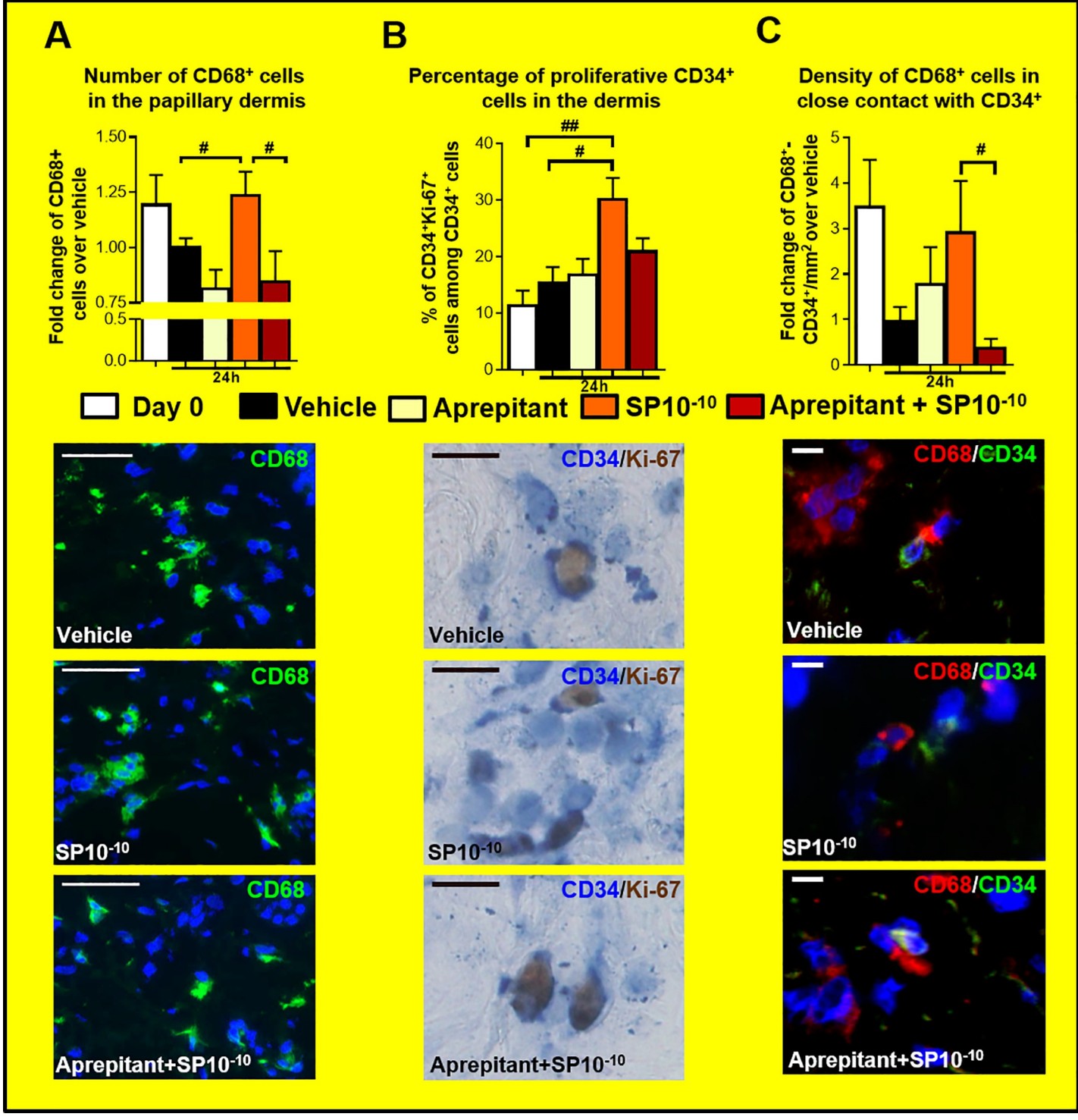

**Fig 6. Aprepitant antagonizes the SP effect on CD34⁺cells and CD68⁺MAC.** (a) Quantitative analysis, and representative images of CD68 immunostainings in human skin fragments at day 0 or treated with vehicle, or SP, or aprepitant, or aprepitant and SP *ex vivo*. The number of single positive cells was counted in the papillary dermis. N = 11–12 skin biopsies/group from 3 donors. Fold change of Mean ± SEM, One-way ANOVA, post hoc test Bonferroni (#p<0.05). Scale bare 50μm (b) Quantitative analysis, and representative images of CD34/Ki-67 immunostainings in human skin fragments at day 0 or treated with vehicle, or SP, or aprepitant, or aprepitant and SP ex vivo. The number of double positive cells was counted in the entire dermis. N = 11–12 skin biopsies/group from 3 donors. Fold change of Mean ± SEM, One-way ANOVA, post hoc test Bonferroni (#p<0.05; ##p<0.01). Scale bare 10μm. (c) Quantitative analysis, and representative images of CD34/CD68 immunostainings in human skin fragments at day 0 or treated with vehicle, or SP, or aprepitant, or aprepitant and SP ex vivo. The number of cells in close contact was counted in the in the papillary dermis. N = 11–12 skin biopsies/group from 3 donors. Fold change of Mean ± SEM, Kruskal-Wallis Test, post hoc test Dunn (#p<0.5). Scale bare 10μm.

induces MAC de novo generation via a direct effect on resident CD34[+]MAC progenitors in human skin. However, alternative indirect mechanisms may also be involved, such as mast cell activation and subsequent histamine release. In fact, SP does not activate mast cells only via NK1R, but also via Mas-related G Protein coupled receptor X2 (MRGPRX2), a receptor that is selectively expressed on mast cells and dorsal root ganglions [34,55]. Once activated by SP, mast cells secrete histamine, and several other cytokines, namely TNFα and IL-33, known to activate and promote MAC maturation [56–58].

Therefore, in this initial pilot study, we have begun to clarify whether SP-induced MAC de novo generation *ex vivo* results from a direct effect on the putative CD34[+]MAC progenitors, or indirect activation of mast cells by administering the selective NK1R antagonist, aprepitant [32–34], so that mast cell activation could be still possible via MRGPRX2.

These additional experiments confirm, once again, that SP stimulation increases significantly MAC number, CD34[+]cell proliferation, and enhanced the co-localization of CD68[+]cells with CD34 cells.

Most importantly, our new results show that the administration of aprepitant 2h before SP stimulation prevented 1) the up-regulation of CD68[+]cells, which number remained comparable to vehicle sample (Fig 6A), 2) the increase in CD34[+]cell proliferation (Fig 6B) and, 3) the stimulation of co-localizations between CD68[+]cells with CD34[+]cells, induced by SP administration (Fig 6C). Thus, our new results demonstrate that NK-1R-mediated signalling is exclusive involved in the SP-induced up-regulation of intradermal macrophage number *ex vivo*, and suggests a direct effect of SP on resident CD34[+]MAC progenitors in human skin: if mast cells were involved in the observed SP effects, they should have been activated by SP via MRGPRX2 [34,55] in the presence of aprepitant, which does not block MRGPRX2 signaling [34].

## Conclusions

Although our immunohistomorphometry-based in situ findings should be systematically followed up by FACS analysis and single cell RNAseq so as to obtain further insights into how SP stimulation impacts on the tissue-resident immunocyte progenitor cells in human skin and their differentiation into CD68[+] macrophages, these intriguing results suggest that CD68[+]MACs can indeed mature from resident intracutaneous CD31[-]CD34[+]mesenchymal stem cells in human skin, which lose CD34 expression upon differentiation. Together with the clinical observation that monocytopenia patient skin has normal MAC numbers and almost no proliferative CD68[+]cells, yet many CD34[+]cells [15], our *ex vivo*-data further support that human dermal CD68[+]trMACs arise from an expanding pool of CD34[+]MAC progenitors after SP stimulation.

Therefore, human skin can generate MACs *de novo* from pre-existing progenitors such as CD34[+]cells, at least under conditions of neurogenic inflammation, rather than from extravasated MOs. This designates dermal MACs as yet another key innate immunity protagonist besides mast cells [17,18], which can be expanded from resident progenitor cells present within human skin. This also raises the question whether the massive dermal MAC increase seen e.g. in lepromatous leprosy, leishmaniasis, granuloma annulare, and tattoo-associated granulomata [59,60] results not only from extravasating MOs, but also from the excessive local MAC maturation from resident, intracutaneous (CD34[+])progenitor cells. If confirmed, this pathological intradermal MAC differentiation process in human skin would deserve to be targeted therapeutically.

## Supporting information

**S1 Text. Supporting material and methods.**
(DOC)

**S1 Fig. Reference area.** Representative pictures of the defined reference areas in the dermis (200μm from the epidermis) used for our analysis. Scale bare 100 μm. D0: day 0.
(TIF)

**S2 Fig. Effect of SP on MHCII⁺cells.** (a) Quantitative analysis, and representative images of MHCII⁺cells in immunofluorescence staining in human skin fragments at day 0 or treated with vehicle, or SP *ex vivo*. The number of cells was counted in the papillary dermis (200μm from the epidermis). N = 11–12 skin biopsies/group from 3 different donors. Fold change of Mean ± SEM, One-way ANOVA, post hoc test Bonferroni (#p<0.05).
(a) Quantitative analysis, and representative images of MHCII⁺CD11c⁺cells in immunofluorescence staining in human skin fragments at day 0 or treated with vehicle, or SP *ex vivo*. The number of double positive cells was counted in the papillary dermis (200μm from the epidermis). N = 11–12 skin biopsies/group from 3 different donors. Fold change of Mean ± SEM, One-way ANOVA, post hoc test Bonferroni (#p<0.05; ##p<0.01). Orange arrows indicate double positive cells.
Scale bare: 50μm.
(TIF)

**S3 Fig. H&E representation of skin capillaries.** Representative pictures showing skin capillaries and blood smear control. Erythrocytes (red arrows) were visualized in few capillaries only at day 0. Intraluminal MO (blue arrow) was detected in a single lumen at day 0. Perivascular MACs (green arrows) were identify at day 0 and Vehicle control. Blood smear control showing erythrocytes and circulating T-cells (black arrows).
(TIF)

**S4 Fig. Proliferative CD68⁺cells.** (a) Representative pictures of the internal positive control showing PH3⁺ cells in the epidermis. Very few PH3⁺CD68⁺MACs were visualized in D0 (day 0). This staining was qualitatively evaluated in 96 sections derived from 4 punches per conditions from 2 different donors. White arrow indicates a double positive CD68⁺PH3⁺cell (white).
(b) Representative pictures of the internal positive control showing EdU⁺ proliferative cells in the epidermis. Very few EdU⁺CD68⁺ MACs were visualized in vehicle and SP 10⁻¹⁰M treated human scalp skin. EdU⁺CD68⁺ cells were detected out of 32 sections derived from 4 punches per conditions from 1 donor. Pink arrow indicates a double positive CD68⁺EdU⁺ cells. EP: epidermis; PD: papillary dermis.
Scale bare 50μm.
(TIF)

**S5 Fig. Characterization of CD34⁺progenitor cells.** (a) Representative picture showing CD34/NK1R cells. Black Arrows indicate double positive CD34⁺NK1R⁺ cells.
(b) Quantitative analysis, and representative images of CD34/CD31 in immunofluorescence staining in human skin fragments at day 0 or treated with vehicle, or SP *ex vivo*. The percentage of double-positive cells was counted in the papillary dermis (200μm from the epidermis). N = 7–8 skin biopsies/group from 2 different donors. Mean ± Men. Green arrows indicate CD34⁺CD31⁻cells. Scale bare: 50μm.
(TIF)

**S1 Table. Primary antibodies employed.** Antibodies used for immunofluorescence stainings are listed and described in detail.
(DOCX)

**S2 Table. List of all the immunostainings.** Immunostaining performed and relevant details. Tris-buffered saline (TBS), phosphate buffered saline (PBS), 4',6-diamidin-2'-phenylindoldi-hydrochlorid (DAPI).
(DOCX)

**S3 Table. List of the secondary antibodies.** Fluorescein isothiocynate (FITC).
(DOCX)

## Acknowledgments

The authors are greatly indebted to our collaborating surgeons, who kindly provided human skin samples, namely to Dr. W. Funk, Munich, and Dr. Meyburg, Berlin. Dr. C. Hillgruber is gratefully acknowledged for helpful experimental advice, and Dr. M. Alam for his help with text editing. This research was initiated while most of the authors worked together at the University of Münster.

## Author Contributions

**Conceptualization:** Jennifer Gherardini, Jérémy Chéret, Kimberly Mace, Marta Bertolini, Ralf Paus.

**Data curation:** Jennifer Gherardini, Youhei Uchida, Jonathan A. Hardman, Jérémy Chéret, Marta Bertolini.

**Formal analysis:** Jennifer Gherardini.

**Funding acquisition:** Ralf Paus.

**Investigation:** Jennifer Gherardini.

**Methodology:** Jennifer Gherardini, Youhei Uchida, Jonathan A. Hardman, Jérémy Chéret.

**Project administration:** Marta Bertolini.

**Resources:** Ralf Paus.

**Software:** Jérémy Chéret.

**Supervision:** Marta Bertolini, Ralf Paus.

**Validation:** Marta Bertolini, Ralf Paus.

**Writing – original draft:** Jennifer Gherardini, Marta Bertolini, Ralf Paus.

**Writing – review & editing:** Jennifer Gherardini, Jonathan A. Hardman, Jérémy Chéret, Kimberly Mace, Marta Bertolini, Ralf Paus.

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
