## [Decision Letter · Decision Letter 0]

11 Sep 2019

PONE-D-19-18895

Tissue-resident macrophages can be generated de novo in adult human skin during neurogenic inflammation from resident progenitor cells

PLOS ONE

Dear Dr. Gherardini,

Thank you for submitting your manuscript to PLOS ONE. After careful consideration, we feel that it has merit but does not fully meet PLOS ONE’s publication criteria as it currently stands. Therefore, we invite you to submit a revised version of the manuscript that addresses the points raised during the review process.

As you will recognize from the comments of the reviewers they raised some points of critique, especially regarding the role of histamin in your findings, characterization of cells and presentation of the manuscript.

We would appreciate receiving your revised manuscript within 2 months. To enhance the reproducibility of your results, we recommend that if applicable you deposit your laboratory protocols in protocols.io, where a protocol can be assigned its own identifier (DOI) such that it can be cited independently in the future. For instructions see: http://journals.plos.org/plosone/s/submission-guidelines#loc-laboratory-protocols

We look forward to receiving your revised manuscript.

Kind regards,

Rudolf Kirchmair

Academic Editor

PLOS ONE

Journal Requirements:

"The work was partially supported by a basic research grant and a PhD fellowship awarded to J.G. from Monasterium Laboratory GmbH, Münster, as well as by the NIHR Manchester Biomedical Research Centre, and by University of Miami start-up funds (R.P.)."

"The author(s) received no specific funding for this work"

Please provide an amended Funding Statement that declares *all* the funding or sources of support received during this specific study (whether external or internal to your organization) as detailed online in our guide for authors at http://journals.plos.org/plosone/s/submit-now

Please state what role the funders took in the study.  If any authors received a salary from any of your funders, please state which authors and which funder. If the funders had no role, please state: "The funders had no role in study design, data collection and analysis, decision to publish, or preparation of the manuscript."

"The authors have declared that no competing interests exist"

We note that one or more of the authors are employed by a commercial company: Monasterium Laboratory GmbH

Reviewers' comments:

Reviewer's Responses to Questions

**Comments to the Author**

1. Is the manuscript technically sound, and do the data support the conclusions?

Reviewer #1: Yes

Reviewer #2: Yes

2. Has the statistical analysis been performed appropriately and rigorously? 

Reviewer #1: Yes

Reviewer #2: Yes

3. Have the authors made all data underlying the findings in their manuscript fully available?

Reviewer #1: Yes

Reviewer #2: Yes

4. Is the manuscript presented in an intelligible fashion and written in standard English?

Reviewer #1: Yes

Reviewer #2: Yes

5. Review Comments to the Author

Reviewer #1: This manuscript considers the possibility that tissue macrophages can be generated in skin from resident progenitor cells when stimulated by substance P. This is potentially a very interesting manuscript that is novel. I have the following points:

1. As substance P appears to be the only neurogenic mediator used, then I suggest the title includes the name of this neuropeptide.

2. Substance P when added exogenously is able to activate mast cells, to release histamine/5-HT which could be the mechanism here. This should be investigated with histamine and 5-HT antagonists. Additionally an NK1 antagonist could be used. Also there are NK1 agonists. Thus there are simple pharmacological experiments which can be done to further this hypothesis and to give a much clearer indication that it is indeed NK1 receptor-dependent.

Reviewer #2: This manuscript describes a very interesting and intriguing mechanism of macrophage de novo generation from CD34+ mesenchymal stem cells in human skin. This observation may have important impacts in the understanding of several disease mechanisms linked with various skin pathologies.

The results presented are convincing, based on 4 different donors, however, some clarifications would be necessary.

1. Since the overall number of MHC class II cells remains unchanged after SP stimulation, but the number of CD68+ cells increased, it means some MHC class II cells were affected by SP and decreased in numbers. Which one? Dendritic cells?

2. It would be informative to clearly state the amount of increase in CD68+ cells upon SP stimulation, which seems to be rather small.

3. In the introduction, authors mentioned that MACs in skin are entirely or partially self-maintained from proliferating tissue-resident MACS, and not from MOs. However, they showed in figure 2 that 70 to 80% of MAC CD68+ cells were CD14+ positive, which means they are from MO origin. One would expect only a very limited proportion of CD14+CD68+ cells compared to CD14-CD68+ if MACs mostly originate from CD14- cells (MSCs or trMACs).

In figure 2, there was a mix-up between CD14-CD68+ and CD14+CD68-, please correct.

4. In figure 2a, how to discriminate between CD14+ MO and CD14+ MACs (as seen in Fig2b)?

Moreover, to confirm that blood was still trapped in capillaries (and has not been washed during skin processing), it would be useful to stain for red blood cells.

5. In figure 2c, how to explain the decrease of P-selectin between Day 0 and the test with the vehicle. My understanding is that the difference between Day 0 and vehicle is the 24h incubation period, more than a potential toxic effect of the vehicle. This means the tissue is deteriorating fast in this culture condition. Same comment about figure 3c. In this figure, how to explain that decreased proliferation is not observed for CD68+ cells (3b), but high for MHCII cells (3c)?

6. In the conclusion part, the mention that the findings were limited by unavailability of large skin samples is not convincing, since large amount of skin can be obtained from plastic surgeries such as abdominoplasties or breast surgeries. Indeed, a FACS analysis would have been highly informative.

6. PLOS authors have the option to publish the peer review history of their article (what does this mean?). If published, this will include your full peer review and any attached files.

Reviewer #1: No

Reviewer #2: Yes: Francois Berthod

---

## [Author Response · Author response to Decision Letter 0]

3 Dec 2019

EDITOR COMMENTS

As you will recognize from the comments of the reviewers they raised some points of critique, especially regarding the role of histamine in your findings, characterization of cells and presentation of the manuscript.

Thank you. Below, we have attempted to address these items of critique along with the addition of substantial new data sets and improvements in the overall data presentation.

REVIEWER COMMENTS

Reviewer #1: This manuscript considers the possibility that tissue macrophages can be generated in skin from resident progenitor cells when stimulated by substance P. This is potentially a very interesting manuscript that is novel. I have the following points:

1. As substance P appears to be the only neurogenic mediator used, then I suggest the title includes the name of this neuropeptide.

Thanks a lot. Yes, this makes perfect sense. We have changed the title accordingly: 

“Tissue-resident macrophages can be generated de novo in adult human skin from resident progenitor cells during substance P-mediated neurogenic inflammation ex vivo”. 

2. Substance P when added exogenously is able to activate mast cells, to release histamine/5-HT which could be the mechanism here. This should be investigated with histamine and 5-HT antagonists. Additionally an NK1 antagonist could be used. Also there are NK1 agonists. Thus there are simple pharmacological experiments which can be done to further this hypothesis and to give a much clearer indication that it is indeed NK1 receptor-dependent.

We agree with the reviewer that mast cell activation, with subsequent histamine release, might be one of the mechanisms through which substance P could, in theory, promote macrophage de novo generation ex vivo. In fact, substance P does not activate mast cells only via the NK1R, but also via Mas-related G Protein coupled receptor X2 (MRGPRX2), a receptor that is selectively expressed on mast cells and dorsal root ganglions (1,2). Once activated by substance P, mast cells secrete histamine, and several other cytokines, namely TNFa and IL-33, known to activate and promote macrophage maturation (3–5). In addition, a plethora of other preformed and newly synthesized agents are released by mast cells upon degranulation. Therefore, histamine is only one of many other candidate mechanisms, if the observed macrophage effects were indeed indirectly mediated and dependent on mast cell degranulation. To dissect such indirect mechanisms is a very complex task and would require an extensive separate study. 

However, our currently available data already tend to support the alternative explanation that substance P exerts direct effects on resident CD34+ macrophage progenitors in human skin, since we have shown that these cells also express NK1R (Fig S4a), and that the increase in macrophage numbers occurs quite rapidly (within 24 hrs). Incidentally, this corresponds well to our previous demonstration that the stimulation of resident human mast cell progenitors located in the connective tissue sheath of human scalp hair follicles ex vivo with the prototypic stress-associated neurohormone, CRH, suffices to induce substantial maturation of these progenitor cells into fully differentiated, functional and degranulating mast cells (6).

Following the reviewer’s excellent suggestion, we have performed 3 new skin organ culture experiments, in which we selectively antagonized only NK1R with aprepitant (7,8). As explained, this still leaves the possibility that mast cells could be activated by substance P via MRGPRX2 (1,2). These additional experiments confirm, once again, that substance P stimulation increases significantly macrophage number, CD34+ cell proliferation, and enhanced the co-localization of CD68+ cells with CD34+ cells. 

Most importantly, our new results show that the administration of aprepitant 2h before substance P stimulation prevented 1) the up-regulation of CD68+ cells, which number remained comparable to vehicle sample, 2) the increase in CD34+ cell proliferation and, 3) the stimulation of co-localizations between CD68+ cells with CD34+ cells, induced by substance P administration (see revised Fig. 6a-c). Thus, our new results demonstrate that NK-1R-mediated signalling is exclusively involved in the substance P induced up-regulation of intradermal macrophage number ex vivo, and suggests a direct effect of substance P on resident CD34+ macrophage progenitors in human skin: if mast cells were involved in the observed substance P effects, they should have been activated by substance P via MRGPRX2 (1,2) in the presence of aprepitant, which does not block MRGPRX2 signalling . 

We have added this data in the revised Results and Discussion session, page 14, and Fig 6a-c.

Reviewer #2: This manuscript describes a very interesting and intriguing mechanism of macrophage de novo generation from CD34+ mesenchymal stem cells in human skin. This observation may have important impacts in the understanding of several disease mechanisms linked with various skin pathologies. The results presented are convincing, based on 4 different donors, however, some clarifications would be necessary.

1. Since the overall number of MHC class II cells remains unchanged after SP stimulation, but the number of CD68+ cells increased, it means some MHC class II cells were affected by SP and decreased in numbers. Which one? Dendritic cells?

Thanks for raising this important point, which we had insufficiently addressed. Indeed, substance P acts also on other MHCII+ immunocytes which are, besides macrophages, namely monocytes (7,8), dendritic cells (7,9) and occasionally mast cells (10). It is known that, under conditions of neurogenic inflammation, dermal dendritic cells respond fast and migrate to the lymph node (11,12). Therefore, given that we did not see a decrease in the number of CD14+ monocytes (Fig 2b), just like the expert referee, we hypothesized that the MHC class II+ cell population decreasing after substance P stimulation are indeed dendritic cells. Given that several subtypes of dendritic cells are present in human dermis and that it is still not entirely clear which is the best marker to use (13,14), we have opted for a double immunostaining protocol of MHC class II with CD11c.

Our new results show that the number of MHCII+CD11c+ dendritic cells is reduced by ca. 20% in SP-treated skin samples compared to vehicle controls. This explains the essentially unaltered number of MHC class II+ cells after SP stimulation, since the decrease in CD11c+ cells is counterbalanced by the observed increase in the number of (also MHCII+) CD68+ macrophages. These new results are now presented and discussed on page 7, and shown in the new Fig S2b.

2. It would be informative to clearly state the amount of increase in CD68+ cells upon SP stimulation, which seems to be rather small.

Done as requested. We now state clearly in the revised Results and Discussion section (p.6) that the difference of CD68+ cell number between samples treated with Substance P 10-10M and vehicle samples is 66.8 cell/mm2 (i.e. an increase of 34% compared to vehicle control). Given that the increase in the number of CD68+ immunocytes consistently occurred within only 24 h of SP stimulation, and this in a non-blood-perfused tissue, this constitutes a remarkable numeric enhancement within a very short time window. This underscores the great potential of human dermis to regenerate macrophages from resident progenitors. We have now emphasized this in the revised manuscript (see Results and Discussion page 6, Fig 1)

3. In the introduction, authors mentioned that MACs in skin are entirely or partially self-maintained from proliferating tissue-resident MACS, and not from MOs. However, they showed in figure 2 that 70 to 80% of MAC CD68+ cells were CD14+ positive, which means they are from MO origin. One would expect only a very limited proportion of CD14+CD68+ cells compared to CD14-CD68+ if MACs mostly originate from CD14- cells (MSCs or trMACs).

Thanks for inviting us to clarify this. The sentence in the introduction “…including skin, MACs are entirely or partially self-maintained from proliferating tissue-resident MACs (trMACs) of embryonal origin” refers to murine tissues, including mouse skin. Indeed, the data of Tamoutounour et al. (2013) in mouse skin demonstrate that around 20% of CD68+ macrophage are Ly-6C- (i.e. the mouse analogue of CD14), indicating the existence of a pool of dermal macrophages that is established prenatally and persists in adulthood (7,8,15), showing that macrophages are indeed “partially” self-maintained by proliferating tissue-resident MACs of embryonal origin. 

Therefore, the fact that we also found around 25% of CD68+ macrophage in human skin to be negative for CD14 is nicely in line with mouse data and suggests that a substantial portion of human dermal macrophages are also maintained independently from circulating monocytes. We have added this interesting and important discussion point in the revised Results and Discussion, see page 8.

In figure 2, there was a mix-up between CD14-CD68+ and CD14+CD68-, please correct.

Thank you very much for spotting this error, which we have corrected. 

4. In figure 2a, how to discriminate between CD14+ MO and CD14+ MACs (as seen in Fig2b)?

Moreover, to confirm that blood was still trapped in capillaries (and has not been washed during skin processing), it would be useful to stain for red blood cells.

The aim of figure 2a is to identify monocytes trapped in the lumen of the blood vessels, that potentially could have extravasated and given rise to dermal macrophages. (Since macrophages are not circulating cells, the chance to find CD14+ macrophage trapped in blood vessels was obviously minimal.) However, we have followed the excellent suggestion of the reviewer and have used H&E to histochemical stain red blood cells and to discriminate macrophages (large irregularly shaped cells) from monocytes (smaller and more rounded cells) (16, 17) in the capillary lumen.

These additional analyses revealed that no MACs could be found to be trapped in the blood vessels of any of the skin samples analysed, confirming that the isolate, few intracapillary CD14+ cells that are found are indeed CD14+ circulating monocytes. 

We have also found very few red blood cells in skin samples before the set-up of organ culture, but not in vehicle- or SP-treated skin samples, i.e. after organ culture. Therefore, while some blood remained trapped in the blood vessels after punches preparation, most of it was washed out during organ culture, further supporting our hypothesis that the newly generated MAC after SP stimulation did not derive from CD14+ circulating monocytes. These new data are now shown in the new Fig. S3 and are correspondingly addressed in the revised Results and Discussion section (p 8, fig S3).

5. In figure 2c, how to explain the decrease of P-selectin between Day 0 and the test with the vehicle. My understanding is that the difference between Day 0 and vehicle is the 24h incubation period, more than a potential toxic effect of the vehicle. This means the tissue is deteriorating fast in this culture condition. Same comment about figure 3c. In this figure, how to explain that decreased proliferation is not observed for CD68+ cells (3b), but high for MHCII cells (3c)?

Although organ-cultured human skin indeed represents a slowly degenerating assay system over time (17,18), it remains fully viable for several days, and most certainly during the short 24 h incubation period chosen here. If it were a fast-deteriorating system most other resident skin cells would have undergone massive apoptosis, which is clearly not the case here. Instead, as the reviewer pointed out, we did not detect a significant downregulation in the proliferation of CD68+ cells (fig 3b), CD34+ cells (fig 4c), or epidermal keratinocytes (fig 3b), and neither saw a significant up-regulation of apoptotic CD68+ cells in the vehicle group after 24 h compared to day 0. Therefore, the decrease in P-selectin expression, or proliferative MHC class II cell number between Day 0 and vehicle cannot be explained as consequence of rapid skin deterioration. 

However, the fact that P-selectin expression is up-regulated in endothelial cells (19,20) of day 0 skin samples as compared to vehicle samples may well result from the pro-inflammatory environment triggered in the skin sample during the manipulation of the skin samples, from the trauma of surgical skin harvesting to the initiation of the culture (day 0) (21). Thus, the expression of P-selectin is most likely restored to the baseline level during organ culture, as the skin gradually adjusts to its new ex vivo environment. Instead, the decrease in the proliferative MHC class II+ cell number detected in vehicle samples compared to day 0 samples may be explained by the well-documented fact that dendritic cells, rather than macrophages (9), rapidly migrate out of the skin into the culture medium under tissue stress conditions (10,11). This has been briefly acknowledged in the Results and Discussion section (p 9 and p 11)

6. In the conclusion part, the mention that the findings were limited by unavailability of large skin samples is not convincing, since large amount of skin can be obtained from plastic surgeries such as abdominoplasties or breast surgeries. Indeed, a FACS analysis would have been highly informative.

While the reviewer is right about the availability of skin from the indicated sources, as a laboratory that specializes in hair research and has a special interest in the role of perifollicular macrophages in scalp skin (22–24), we purposely used small residual skin fragments derived from face-lift surgery, which severely limited the amount of available human skin for organ culture. This rationale is now explained more clearly in the revised Material and Methods section (p. 4). We fully agree with expert referee that FACS analysis would have very nicely complemented the current study. We now acknowledge in the conclusion that the pointers provided by the current, quantitative immunohistomorphometry-based study should be followed-up by complementing with FACS analyses (p 16).

References 

1. Azimi E, Reddy VB, Shade K-TC, Anthony RM, Talbot S, Pereira PJS, et al. Dual action of neurokinin-1 antagonists on Mas-related GPCRs. JCI Insight. 2016 Oct 6;1(16):e89362. 

2. Green DP, Limjunyawong N, Gour N, Pundir P, Dong X. A Mast-Cell-Specific Receptor Mediates Neurogenic Inflammation and Pain. Neuron. 2019 06;101(3):412-420.e3. 

3. Varricchi G, Rossi FW, Galdiero MR, Granata F, Criscuolo G, Spadaro G, et al. Physiological Roles of Mast Cells: Collegium Internationale Allergologicum Update 2019. Int Arch Allergy Immunol. 2019;179(4):247–61. 

4. Jackson-Jones LH, Rückerl D, Svedberg F, Duncan S, Maizels RM, Sutherland TE, et al. IL-33 delivery induces serous cavity macrophage proliferation independent of interleukin-4 receptor alpha. Eur J Immunol. 2016 Oct;46(10):2311–21. 

5. Murray PJ, Allen JE, Biswas SK, Fisher EA, Gilroy DW, Goerdt S, et al. Macrophage Activation and Polarization: Nomenclature and Experimental Guidelines. Immunity. 2014 Jul 17;41(1):14–20. 

6. Ito N, Sugawara K, Bodó E, Takigawa M, van Beek N, Ito T, et al. Corticotropin-releasing hormone stimulates the in situ generation of mast cells from precursors in the human hair follicle mesenchyme. J Invest Dermatol. 2010;130(4):995–1004. 

7. Haniffa M, Ginhoux F, Wang X-N, Bigley V, Abel M, Dimmick I, et al. Differential rates of replacement of human dermal dendritic cells and macrophages during hematopoietic stem cell transplantation. J Exp Med. 2009 Feb 16;206(2):371–85. 

8. McGovern N, Schlitzer A, Gunawan M, Jardine L, Shin A, Poyner E, et al. Human Dermal CD14+ Cells Are a Transient Population of Monocyte-Derived Macrophages. Immunity. 2014 Sep 18;41(3):465–77. 

9. Toebak MJ, Gibbs S, Bruynzeel DP, Scheper RJ, Rustemeyer T. Dendritic cells: biology of the skin. Contact Dermatitis. 2009;60(1):2–20. 

10. Kambayashi T, Allenspach EJ, Chang JT, Zou T, Shoag JE, Reiner SL, et al. Inducible MHC Class II Expression by Mast Cells Supports Effector and Regulatory T Cell Activation. J Immunol. 2009 Apr 15;182(8):4686–95. 

11. Joachim RA, Handjiski B, Blois SM, Hagen E, Paus R, Arck PC. Stress-Induced Neurogenic Inflammation in Murine Skin Skews Dendritic Cells Towards Maturation and Migration: Key Role of Intercellular Adhesion Molecule-1/Leukocyte Function-Associated Antigen Interactions. Am J Pathol. 2008 Nov 1;173(5):1379–88. 

12. Arck PC, Handjiski B, Peters EMJ, Peter AS, Hagen E, Fischer A, et al. Stress inhibits hair growth in mice by induction of premature catagen development and deleterious perifollicular inflammatory events via neuropeptide substance P-dependent pathways. Am J Pathol. 2003 Mar;162(3):803–14. 

13. Zaba LC, Krueger JG, Lowes MA. Resident and “Inflammatory” Dendritic Cells in Human Skin. J Invest Dermatol. 2009 Feb 1;129(2):302–8. 

14. Clark GJ, Silveira PA, Hogarth PM, Hart DNJ. The cell surface phenotype of human dendritic cells. Semin Cell Dev Biol. 2019 Feb 1;86:3–14. 

15. Tamoutounour S, Guilliams M, Montanana Sanchis F, Liu H, Terhorst D, Malosse C, et al. Origins and functional specialization of macrophages and of conventional and monocyte-derived dendritic cells in mouse skin. Immunity. 2013 Nov 14;39(5):925–38. 

16. Burke B, Lewis CE. The Macrophage. Second edition. Oxford ; New York: Oxford University Press; 2002. 680 p. 

17. Lu Z, Hasse S, Bodo E, Rose C, Funk W, Paus R. Towards the development of a simplified long-term organ culture method for human scalp skin and its appendages under serum-free conditions. Exp Dermatol. 2007 Jan;16(1):37–44. 

18. Zhou L, Zhang X, Paus R, Lu Z. The renaissance of human skin organ culture: A critical reappraisal. Differ Res Biol Divers. 2018 Dec;104:22–35. 

19. Gerhardt T, Ley K. Monocyte trafficking across the vessel wall. Cardiovasc Res. 2015 Aug 1;107(3):321–30. 

20. Vestweber D. How leukocytes cross the vascular endothelium. Nat Rev Immunol. 2015 Nov;15(11):692–704. 

21. Uchida Y, Gherardini J, Alam M, Keren A, Zhang H, Chéret J, et al. Human dermal Vδ1+T-cells recognize “stressed” HFs and may induce alopecia areata. J Dermatol Sci. 2017 May 1;86(2):e59. 

22. Hardman JA, Muneeb F, Pople J, Bhogal R, Shahmalak A, Paus R. Human perifollicular macrophages undergo apoptosis, express Wnt ligands and switch their polarisation during catagen. J Invest Dermatol. 2019 Jun 21; 

23. Christoph T, Müller-Röver S, Audring H, Tobin DJ, Hermes B, Cotsarelis G, et al. The human hair follicle immune system: cellular composition and immune privilege. Br J Dermatol. 2000;142(5):862–873. 

24. Muneeb F, Hardman JA, Paus R. Hair growth control by innate immunocytes: Perifollicular macrophages revisited. Exp Dermatol. 2019 Apr;28(4):425–31.

---

## [Decision Letter · Decision Letter 1]

31 Dec 2019

Tissue-resident macrophages can be generated de novo in adult human skin from resident progenitor cells during substance P-mediated neurogenic inflammation ex vivo.

PONE-D-19-18895R1

Dear Dr. Gherardini,

We are pleased to inform you that your manuscript has been judged scientifically suitable for publication and will be formally accepted for publication once it complies with all outstanding technical requirements.

With kind regards,

Rudolf Kirchmair

Academic Editor

PLOS ONE

Additional Editor Comments (optional):

Reviewers' comments:

Reviewer's Responses to Questions

**Comments to the Author**

1. If the authors have adequately addressed your comments raised in a previous round of review and you feel that this manuscript is now acceptable for publication, you may indicate that here to bypass the “Comments to the Author” section, enter your conflict of interest statement in the “Confidential to Editor” section, and submit your "Accept" recommendation.

Reviewer #1: All comments have been addressed

2. Is the manuscript technically sound, and do the data support the conclusions?

Reviewer #1: Yes

3. Has the statistical analysis been performed appropriately and rigorously? 

Reviewer #1: Yes

4. Have the authors made all data underlying the findings in their manuscript fully available?

Reviewer #1: Yes

5. Is the manuscript presented in an intelligible fashion and written in standard English?

Reviewer #1: Yes

6. Review Comments to the Author

Reviewer #1: (No Response)

7. PLOS authors have the option to publish the peer review history of their article (what does this mean?). If published, this will include your full peer review and any attached files.

Reviewer #1: No

---

## [Editor Report · Acceptance letter]

14 Jan 2020

PONE-D-19-18895R1 

Tissue-resident macrophages can be generated de novo in adult human skin from resident progenitor cells during substance P-mediated neurogenic inflammation ex vivo. 

Dear Dr. Gherardini:

I am pleased to inform you that your manuscript has been deemed suitable for publication in PLOS ONE. Congratulations! Your manuscript is now with our production department. 

With kind regards,

on behalf of

Prof Rudolf Kirchmair 

Academic Editor

PLOS ONE